# SSoC: Learning Spontaneous and Self-Organizing Communication for Multi-Agent Collaboration

## Abstract

Multi-agent collaboration is required by numerous real-world problems. Although distributed setting is usually adopted by practical systems, local range communication and information aggregation still matter in fulfilling complex tasks. For multi-agent reinforcement learning, many previous studies have been dedicated to design an effective communication architecture. However, existing models usually suffer from an ossified communication structure, e.g., most of them predefine a particular communication mode by specifying a fixed time frequency and spatial scope for agents to communicate regardless of necessity. Such design is incapable of dealing with multi-agent scenarios that are capricious and complicated, especially when only partial information is available. Motivated by this, we argue that the solution is to build a spontaneous and self-organizing communication (SSoC) learning scheme. By treating the communication behaviour as an explicit action, SSoC learns to organize communication in an effective and efficient way. Particularly, it enables each agent to spontaneously decide when and who to send messages based on its observed states. In this way, a dynamic inter-agent communication channel is established in an online and self-organizing manner. The agents also learn how to adaptively aggregate the received messages and its own hidden states to execute actions. Various experiments have been conducted to demonstrate that SSoC really learns intelligent message passing among agents located far apart. With such agile communications, we observe that effective collaboration tactics emerge which have not been mastered by the compared baselines.

## 1 Introduction

Many real-world applications involve participation of multiple agents, for example, multi-robot control[13], network packet delivery[21] and autonomous vehicles planning [1], etc.. Learning such systems is ideally required to be autonomous (e.g., using reinforcement learning). Recently, with the rise of deep learning, deep reinforcement learning (RL) has demonstrated many exciting results in several challenging scenarios e.g. robotic manipulation [4][10], visual navigation [23][11], as well as the well-known application in game playing [14][17] etc.. However, unlike its success in solving single-agent tasks, deep RL still faces many challenges in solving multi-agent learning scenarios.

Modeling multiple agents has two extreme solutions: one is treating all agents as an unity to apply a single centralized framework, the other is modelling the agents as completely independent learners. Studies following the former design are often known as "centralized approach", for example [19][15] etc. The obvious advantage of this class of approaches is a good guarantee of optimality since it is equivalent to the single agent Markov decision process (MDP) essentially. However, it is usually unfeasible to assume a global controller that knows everything about the environment in practice. The other class of methods can be marked as "independent multi-agent reinforcement learning". These approaches assumes a totally independent setting in which the agents treat all others as a part of the observed environment. [3] has pointed out that such a setup will suffer from the problem of non-stationarity, which renders it hard to learn an optimal joint policy.

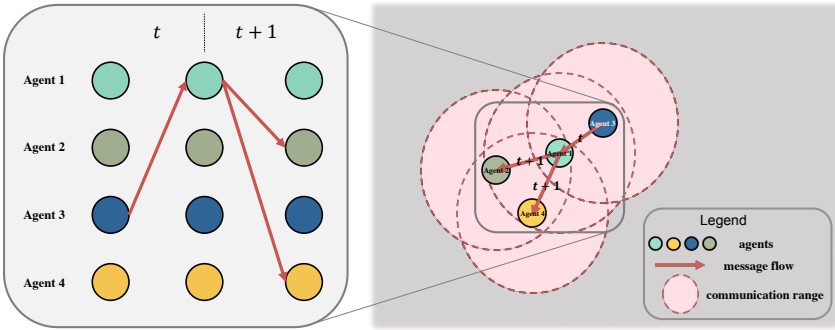

Figure 1: An example to demonstrate the message flow of spontaneous and self-organizing communication (SSoC) architecture on a multi-agent system. The communication path is not predestined. Instead, it is established by a "Speak" action learned by the agents. Each agent can either start a communication or transfer its received messages to remote agents.

Therefore, an intermediate setting of the above two scenarios is adopted by many researches in this field. Specifically, each agent is constrained to a local view of the whole observation, but it can exploit the information from neighbor agents within a local range around it. This paradigm of MARL fits practical cases much better. Several recent efforts make a good progress under such a distributed setup. One group of studies exploit a "centralized/communicated training + decentralized/independent execution" mode. These works include [6][12] etc.. The information sharing is fulfilled by learning a joint value function which exploits all agents' information. Hence their goal is to train each agent's model better via a centralized training. However, this may miss the benefit from an explicit local communication at execution time, which we believe is vital for better collaborations.

The other class of methods move forward along a communication route. Among them, Meanfield [20] proposes to learn collaborations between each agent with a "mean" agent of its neighbors. This encourages information sharing within a local range, however, may still be limited since the communication is through a plain mean action-value function without any information processing or weighting mechanism. Hence the importance of each agent is treated equally by mean-field MARL. This simplification may greatly limit the richness of communication and thus hinder the learning of complex collaborations. Like Meanfield, Commnet [18] also assumes an average message for inter-agent communications. It computes a mean value of the collected information from local teammates, then broadcasts it as a message input to all nearby agents.

In essence, there are three key factors that determine a communication. That is when, where and how the participants initiate the communication. Most of existing approaches, including the above-mentioned Meanfield and Commnet, try to predefine each ingredient and thus lead to an inflexible communication architecture. Recently, VAIN[5] and ATOC[7] incorporate attentional communication for collaborative multi-agent reinforcement learning. Compared with Meanfield and Commnet, VAIN and ATOC have made one step further towards more flexible communication. However, the step is still limited. Take ATOC as an example, although it learns a dynamic attention to diversify agent messages, the message flow is only limited to the local range. This is unfavorable for learning complex and long range communications. The communication time is also specified manually (every ten steps). Hence it is requisite to find a new method that allows more flexible communication on both learnable time and scopes.

In this regard, we propose a new solution with learnable spontaneous communication behaviours and self-organizing message flow among agents. The proposed architecture is named as "Spontaneous and Self-Organizing Communication" (SSoC) network. The key to such a spontaneous communication lies in the design that the communication is treated as an action to be learned in a reinforcement manner. The corresponding action is called "Speak". Each agent is eligible to take such an action based on its current observation. Once an agent decides to "Speak", it sends a message to partners within the communication scope. In the next step, agents receiving this message will decide whether

to pass the message forward to more distant agents or keep silence. This is exactly how SSoC distinguishes itself from existing approaches. Instead of predesting when and who will participate in the communication, SSoC agents start communication only when necessary and stop transferring received messages if they are useless. A self-organizing communication policy is learned via maximizing the total collaborative reward. The communication process of SSoC is depicted in Fig.1. It shows an example of the message flow among four communicating agents. Specifically, agent 3 sends a message to ask for help for remote partners. Due to agent 3's communication range, the message can be seen only by agent 1. Then agent 1 decides to transfer the collected message to its neighbors. Finally agent 2 and agent 4 read the messages from agent 3. These two agents are directly unreachable from agent 3. In this way, each agent learns to send or transfer messages spontaneously and finally form a communication route. Compared with the communication channels predefined in previous works, the communication here is dynamically changing according to real needs of the participating agents. Hence the communication manner forms a self-organizing mechanism.

We instantiate SSoC with a policy network with four functional units as shown in Fig.2. Besides the agent's original action, an extra "Speak" action is output based on the current observation and hidden states. Here we simply design "Speak" as a binary $\{0, 1\}$ output. Hence it works as a "switch" to control whether to send or transfer a message. The "Speak" action determines when and who to communicate in a fully spontaneous manner. A communication structure will naturally emerge after several steps of message propagation. Here in our SSoC method, the "Speak" policy is learned by a reward-driven reinforcement learning algorithm. The assumption is that a better message propagation strategy should also lead to a higher accumulated reward.

We evaluate SSoC on several representative benchmarks. As we have observed, the learned policy does demonstrate novel clear message propagation patterns which enable complex collaborative strategies, for example, remote partners can be requested to help the current agent to get over hard times. We also show the high efficiency of communication by visualizing a heat map showing how often the agents "speak". The communication turns out to be much sparser than existing predefined communication channels which produce excessive messages. With such emerged collaborations enabled by SSoC's intelligent communication manner, it is also expected to see clear performance gains compared with existing methods on the tested tasks.

## 2 RELATED WORK

Recently, several studies have concentrated on learning multiple agent communication for deep RL networks. Among them, one class of work tries to build pairwise structure for multi-agent communication. [2] and [9] are among the first to propose learnable communications via back-propagation between individual deep Q-networks. However, due to their motivating tasks, both works rely on a peer-to-peer (P2P) communication strategy and usually apply to only a limited number of agents. VAIN[5] also builds a pair-wise communication structure whose interaction weights are learned. [16] proposes a pair-wise MARL approach for multi-object tracking task. In such models, the essential state-action space of multiple agents grows geometrically with the number of agents, and so is the communication load. Hence the pair-wise models suffer from scale issues.

Another branch of study attempts to establish a global communication channel for all agents. For example, [15] proposes a bidirectional communication channel among all agents to facilitate effective communication. [19] applies a centralized method with zero-order optimization to control all agents. [8] uses a master-slave architecture to control the communication of local agents by a master. Commnet[18] builds a broadcasting communication channel among all agents and mean hidden states are directly used as messages. Such architectures are practical communication ways but still limited since 1) such a predefined structure forces every agent to send messages at every step. This will produce lots of redundant information; 2) They all use a single net to model all agents, which makes it difficult to extend to large-scale tasks.

To address the large-scale multi-agent problem in MARL, [20] proposes Meanfield algorithm. It models a local communication between each agent and an approximating "mean-agent" of all its neighbors. This encourages information sharing within a local range, however, its capability is still limited since the average of action-value function may not be rich enough to induce complex collaborative policies for complicated tasks. Several recent studies exploit a "centralized/communicated training + decentralized/independent execution" mode. For example, in COMA[6], a global critic

was proposed, which could potentially work at a centralized level, however since critics are basically value networks, they do not provide explicit policy guidance. [12] also presents an adaptation of actor-critic method which uses centralized action-value function that takes as input the states and inferred actions of all agents. However, this mainly fosters a better training of each independent model instead of establishing an explicit communication and information processing module at testing time which is essential for effective collaborations

In order to learn more effective communication, [7] uses a recurrent attention model to help dynamically perform communication and have better local communication. However, its message flow is only limited to the local range. This is unfavorable for learning complex and long range communications. The communication time is also specified manually (every ten steps). Compared with their work, SSoC learns both when and who to communicate with a spontaneous communication action. It also allows long range communication via multi-step message propagation. These characteristics are critical for learning better collaborative policies.

## 3 Approach

Here we introduce the details of the proposed SSoC network. We assume a distributed partially observable MARL environment for SSoC. As illustrated by Fig.1, each agent can only see a neighboring circular area. The goal is to achieve a certain winning state via taking a sequence of actions. At each time $t$, an agent takes an action $a_t$ based on its own observed state $s_t$ as well as information sent by partner agents within a local range around it. In this paper we call the shared information $m_t^i$ as "messages". The messages flowing among agents determine which agents are sharing information with others at a certain time step. Unlike previous methods which usually predestine a static communication structure, SSoC adopts a more flexible option. The key is to enable each agent to learn an extra "Speak" action to determine whether to send the message to neighboring agents. Hence only those agents "Speak" currently are able to be "heard" by its partners within its communication range. Then these partner agents are able to exploit the received messages to determine their own actions. They are also eligible to pass the received messages forward to its neighbors if they also take a "Speak" action. And their own "thought" can also be merged into the output message. Like ordinary actions, "Speak" action also takes one time step. A communication path will form naturally among all agents through multiple steps of communication. SSoC manages to establish a "global" communication channel in such a self-organizing manner. Unlike previous communication architectures, this "global" communication channel of SSoC is also applicable for a distributed MARL scenario.

### 3.1 The spontaneous "Speak" action

The "Speak" action is generated by an independent output of the network, which is parallel to the original action output of each agent. We design "Speak" action as a binary signal to control whether a message will be passed. Hence the action space of "Speak" contains 2 actions: "1" represents "Speak", "0" stands for "keep silent". This binary "Speak" signal will be multiplied with the output message of the agent. If the signal is "1", the output message is preserved and will be accessible by the neighbor agents' LWU module. Otherwise, it is hidden from other agents by a 0 signal.

Since the "Speak" action is determined by the agent itself based on its own observation, the subsequent communication is started in a spontaneous way. It may happen anytime, anywhere which only depends on the agents' own policy. It is neither predetermined nor manually designed with human intuitions. Therefore, we name such kind of communication as "self-organizing". Here we re-check the three key factors of multi-agent communication mentioned in section 1. Obviously, such a "self-organizing" communication of SSoC is capable of learning when and who to communicate simultaneously. As for "how to communicate", each agent is able to combine the received message with its own "thought" to output actions. Furthermore, since SSoC implicitly builds a "global" communication channel based on multi-step of "Speak" actions, the communication is not only limited to the local area. Therefore, SSoC's self-organizing communication really addresses the key challenges of multi-agent communication in a new and more flexible way.

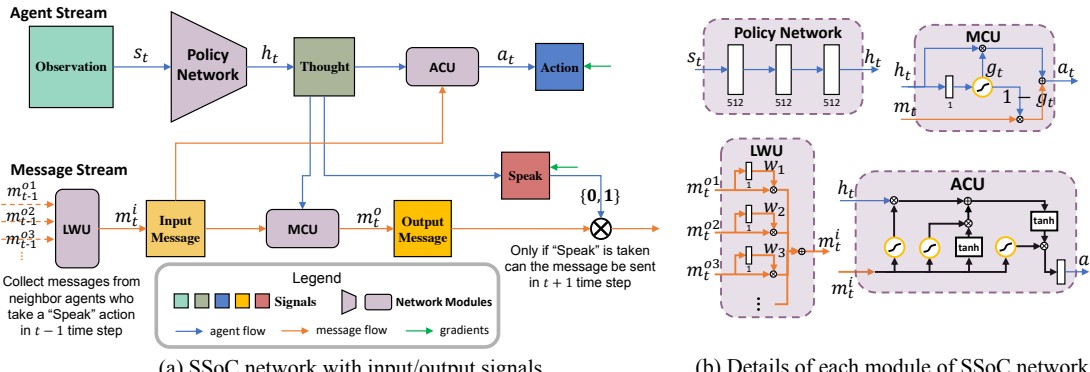

(a) SSoC network with input/output signals     (b) Details of each module of SSoC network

Figure 2: Structure of SSoC network.

## 3.2 NETWORK STRUCTURE

The whole structure of SSoC is displayed as Fig.2 (a). The network mainly consists of two streams. The top "agent stream" of Fig.2 (a) shows the original policy network of each agent. By taking the input observation, a policy network will generate a hidden state "thought" $h_t$. Then the "thought" $h_t$ will be merged via an ACU (action composition unit) module with messages $m_t^i$ from the bottom "message stream" to obtain the action $a_t$. And an extra binary "Speak" action will also be generated from $h_t$ to decide whether to send the agent's message to others. The message stream works as a message receiver & sender for the agent. At each time step, it aggregates messages from other agents within the communication range via a LWU (learnable weighting unit). Only messages from agents who "Speak" at last time step will get collected by LWU. To incorporate the agent's own "thought" and send it to other agents, the aggregated messages $m_t^i$ will be combined with $h_t$ through a MCU (message composition unit) module as the output message $m_t^o$. Finally, the binary "Speak" action from agent stream will be applied to determine the sending of message $m_t^o$ to other agents.

Fig.2 (b) gives detailed implementation of the four functional units of SSoC. Specifically, the policy network consists of 3 fully-connected layers The LWU module works as a self-weighted aggregator of the collected messages $\{m_t^{o1}, m_t^{o2}, m_t^{o3}, \ldots\}$ from neighbor agents. For each input message, it will predict a weight which will be multiplied with the original signal before a sum operation with other messages. MCU merges the agent's own "thought" and input message as a new message that will be spread to other agents. It adopts a learnable gating mechanism which enables the agent to decide the proportion of its own "thought" and other agents' information in its output message. Like MCU, ACU also combines the input message with its own "thought". However, its goal is to output the agent's action. It is implemented like a basic LSTM cell without temporal recurrence.

## 3.3 LEARNING

The agent's own action will be updated by a regular policy gradient with baseline. And the "Speak" action takes an additional "speak" policy gradient as equation 1:

$$\nabla_\theta J_{speak}(\theta) = \nabla_\theta \left[ \log \pi_\theta \left( \boldsymbol{s}_t, \boldsymbol{a}_t^{speak} \right) \left( \sum_{i=t}^{T} r_i - b(\boldsymbol{s}_t, \theta) \right) \right] \tag{1}$$

where $T$ is the total time step of the episode. These two gradients will be fed to SSoC network respectively for back-propagation (as shown by the green arrows in Fig.2 (a)). Note that the output messages will be re-used as the input of other agents in next time step. Hence we create a buffer to store the messages $m_t^o$ at each time step. During training, messages will be taken out by each agent from this buffer for both feed-forward sampling and backward updates. The detailed training procedure is given in the appendix.

## 4 EXPERIMENTS

Experiments are conducted on multi-camera intelligent surveillance task which is firstly proposed by this paper and the large-scale battle task (64 vs. 64) from the MAgent benchmark[22]. Both tasks assume distributed collaborative MARL environments. Agents only have local observations and take actions independently. A shared reward will be given to all agents for model training. For comparison, we select two state-of-the-art methods [18][20] as well as the independent multi-agent policy gradient algorithm (denoted as Independent) as our baselines.

### 4.1 MULTI-CAMERA INTELLIGENT SURVEILLANCE TASK

In real life there are many occasions that need multiple agents cooperation to achieve their common goal. One interesting case is to make a set of intelligent cameras track target people in airport or railway station by actively changing their pose. Each camera is manipulated by an agent. Nearby agents can share their information with each other. Inspired by this scenario, we design a simulation task which we name as multi-camera intelligent surveillance task (as illustrated in Fig.3). In this task, each target person (simplified as the small red balls) has his/her own walking route. They will appear in the entrance area randomly at the beginning and walk to the destination. There are three self-controlling cameras (the blue sector represents the camera's current visual field) trying to actively capture the moving target balls as long as possible. The camera can choose to sway its angle left, right (by 15 degree) or keep static. Each camera can share its own "thought" only with its neighbor camera. Hence this is a distributed MARL setting.

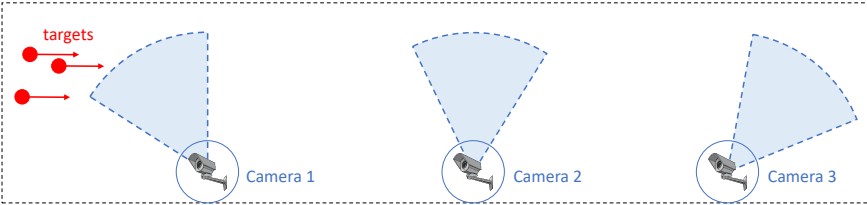

Figure 3: An illustration of the multi-camera intelligent surveillance task.

The general objective of the cameras is to capture the moving balls as long as possible by changing their angles cooperatively. We formulate the multi-camera intelligent surveillance task as a multi-agent reinforcement learning problem:

- Overall objective: tracking the target red balls as long as it can during the whole episode;
- State: the sector visual field's position and the ball's position (given only if it is within the visual field, otherwise is empty);
- Action: turning left by 15 degree, turning right by 15 degree or stay static;
- Reward: if the camera changes its angle and captures the target ball, it receives a positive reward; if the ball is in the camera's possible visual field but the camera doesn't capture it, the agent receives a negative reward; Otherwise, it receives a zero reward. (The details of reward definition is given in appendix.)

Next we will show how our SSoC works in this environment. We argue that the camera agent knows nothing about ball's information, it needs to learn the useful information based its local observation by itself. In our algorithm, the agent will adjust and optimize it's policy with training and learn to dynamically send necessary information to its local partners. As shown in Fig.4 (a), when the ball is captured by the camera, it chooses to speak "1" meaning that it has collected the ball's information and send the integrated message to the next camera to help the next camera learning better. When the ball is outside of all the cameras' field, the cameras choose to speak "0" meaning that there is no useful and necessary message to be delivered between cameras.

We compare SSoC on the multi-camera intelligent surveillance task with Meanfield, Commnet and Independent MARL. In our environment setting, each episode contains 30 time steps And we train every algorithm for 5000 episodes in total. We show the results of SSoC with baseline algorithms

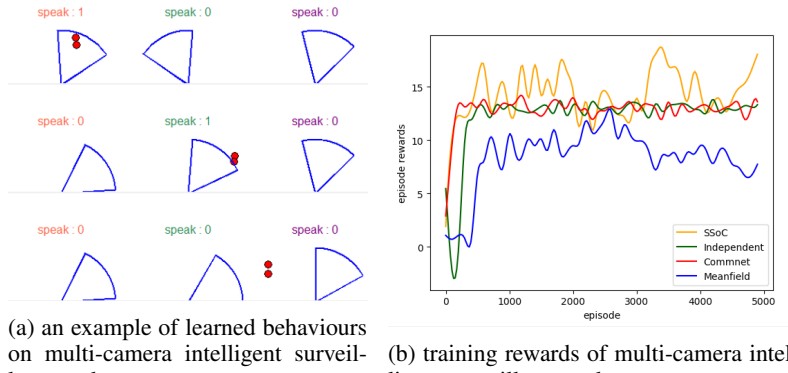

(a) an example of learned behaviours on multi-camera intelligent surveillance task

(b) training rewards of multi-camera intelligent surveillance task

Figure 4: The experiment of multi-camera intelligent surveillance task

in Fig.4 (b). SSoC performs best by getting highest rewards on average, as in table 1. We find that the agents learn a smart strategy that once the agent captures the ball, it can capture the ball all the time until the ball moves outside its visual field. In addition, once an agent captures a ball, its neighbor agent will search around by turning back and forth to capture the coming ball earlier. This collaborative strategy is enabled by useful messages passing among the agents. When necessary, the agent learns to speak "1" meaning that it decides to transfer the useful message to next agent. When there is no effective message, the agent chooses to be silent. For better illustration, we visualize SSoC's learned policy in the video against other algorithms. The video is in demo. We recommend readers to have a look at the video for better understanding of the policy learned by SSoC.

The comparison of mean rewards is given by table 1.

|  | Commnet | Meanfield | Independent | SSoC |
|---|---|---|---|---|
| Mean rewards | 12.77 | 8.35 | 12.17 | 14.18 |

Table 1: mean rewards on multi-camera intelligent surveillance task

## 4.2 LARGE-SCALE BATTLE TASK

The large-scale battle task is one MARL scenario included in the MAgent environment[22]. In the task our algorithm controls a group of agents to eliminate the opponents. Each agent can move by one cell or attack one nearby enemy at one time step. This task is challenging due to a large number of participating agents (here 64 in our experiments). The difficulty is even larger considering the environment's distributed setting and agents' partial visibility. We adopt a self-play training for all the methods following [22]. All methods are trained for 2000 episodes. Each episode contains 400 steps. Learning rate is set to 0.0001 for all.

SSoC outperforms the compared baselines on mean rewards and mean kill (average number of killed enemies in an episode) with a clear margin as shown by table 2. We also let SSoC play against Commnet, Meanfield and Independent in this task. The results are shown in 6 (b). As we can see, SSoC obtains a higher win rate compared with the baselines. The results demonstrate SSoC is an competitive architecture for distributed MARL tasks.

To verify if it is SSoC's spontaneous and self-organizing communication that brings such a big improvement, we draw the heatmap of communication in Fig.5. The heat value here is computed as an accumulation of the "Speak" signal (1 for "Speak" and 0 for silence) of each cell in recent 5 frames. In this way, we discover several cases with meaningful message flow which help our agents win the battle. One such case is shown in Fig.5. Here we display both the heatmap and corresponding situation in $10^{th}$, $15^{th}$ and $20^{th}$ frame here to show the learned agent policies and message flow. In this case, frontier agents (marked as group a) encounter enemies first and needs support of the agents of group b. Hence they start communication by taking "Speak" actions at $10^{th}$ frame. The messages sent by these agents transfer to group b through several steps. At $15^{th}$ frame, group b agents receive the message and "Speak" to gather nearby agents to move right and help agents on the right. At $20^{th}$ step, group b agents begin to confront enemies and join the attack.

After a while, two groups of agents join force together to eliminate most of the enemies. This example shows that the learned self-organizing communication does help our agents to learn a better collaborative policy. Since the proposed scheme enables multi-step message transferring, an agent's message can be spread to agents on a much further area. Hence such a long-range reinforcement policy can be learned in this case. In addition, we can see for most agents, "Speak" action is not taken. This shows that agents only start a communication when necessary, instead of keeping sending messages all the time like in [18][15]. This phenomenon shows the high efficiency of SSoC communication.

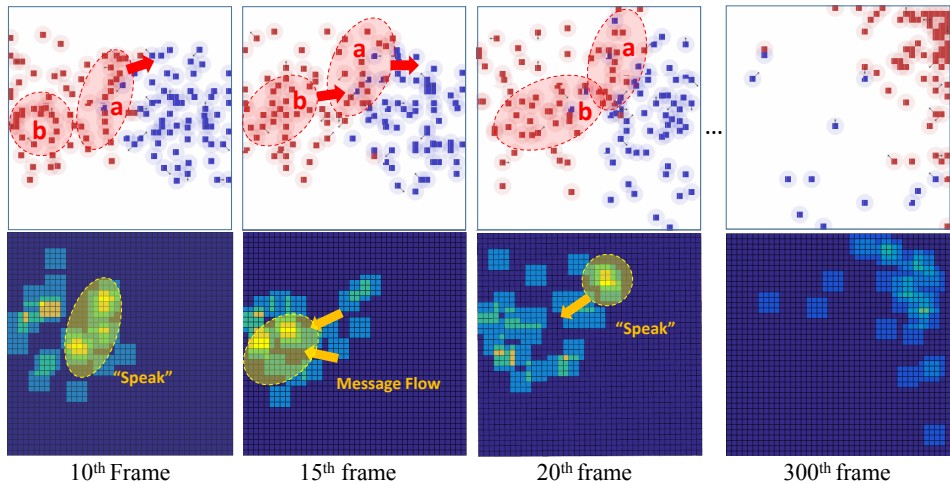

| 10th Frame | 15th frame | 20th frame | 300th frame |

Figure 5: Emerged collaboration eabled by self-organizing communication on one test running.

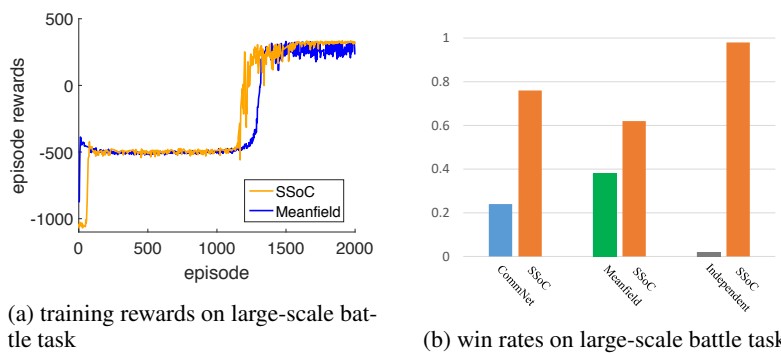

(a) training rewards on large-scale battle task

(b) win rates on large-scale battle task

Figure 6: Performance comparison on large-scale battle task

|  | Commnet | Meanfield | Independent | SSoC |
|---|---|---|---|---|
| Mean rewards | 197.6 | 236.3 | 25.6 | 312.4 |
| Mean Kill | 49 | 52 | 36 | 57 |

Table 2: mean rewards on larget-scale battle task

## 5 CONCLUSION

In this paper, we propose a SSoC network for MARL tasks. Unlike previous methods which often assume a predestined communication structure, the SSoC agent learns when to start a communication or transfer its received message via a novel "Speak" action. Similar to the agent's original action, this "Speak" can also be learned in a reinforcement manner. With such a spontaneous communication action, SSoC is able to establish a dynamic self-organizing communication structure according to the current state. Experiments have been performed to demonstrate better collaborative policies and improved on communication efficiency brought by such a design. In future work, we will continue to enhance the learning of "Speak" action e.g. encoding a temporal abstraction to make the communication flow more stable or develop a specific reward for this "Speak" action.

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

# Appendix

## 1. Reward definition for multi-camera intelligent surveillance task

The reward of the task is designed as follows: if the ball is in area I, the corresponding camera receives a positive reward which is calculated by an constant $c$ subtracting the distance from the target ball to the median of the sector; if the ball is in area II, the camera receives a negative reward which is calculated by the negative of distance from the target ball to the boundary of sector;if the ball is in area III, the camera receives reward of zero. The reward definition is summarized as (1) shows:

$$Reward = \begin{cases} c - d_{in}, & when\ the\ ball\ is\ in\ \text{I} \\ 0, & when\ the\ ball\ is\ in\ \text{II} \\ -d_{out}, & when\ the\ ball\ is\ in\ \text{III} \end{cases} \tag{1}$$

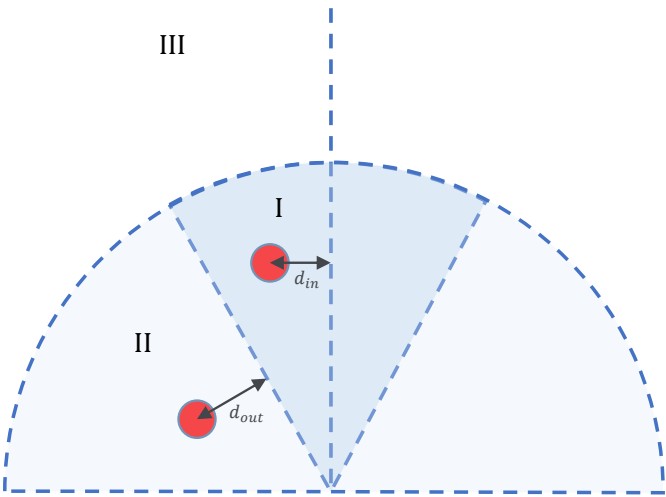

Figure 1: Illustrate the reward defined for multi-camera intelligent surveillance task

## 2. Training details of SSoC on large-scale battle task

The following algorithm gives the process of one round training on the large-scale battle task. The reward is provided by the MAgent[22] platform.

$$\nabla_\theta J(\theta) = \nabla_\theta \left[ \log \pi_\theta \left( \boldsymbol{s}_t, \boldsymbol{a}_t \right) \left( \sum_{i=t}^{T} r_i - b(\boldsymbol{s}_t, \theta) \right) \right] \tag{2}$$

---

**Algorithm 1:** SSoC Multi-Agent Policy Gradient (One Round)

---

Randomly initialize $\theta = (\theta^1, \dots, \theta^n)$;
Set $R_i = 0$, $t = 0$, $T = 400$;
Set $s_1 =$ initial state;
Set $M_{buffer} = \varnothing$
**while** $s_t \neq terminal$ **and** $t < T$ **do**
     $t = t + 1$;
     **for** *each agent $i = 1 \dots n$* **do**
         Observe local state of agent $i$: $s_t^i$;
         Take out messages from $M_{buffer}$ as $M_t = \{m_{t-1}^{o1}, m_{t-1}^{o2}, m_{t-1}^{o3}, \dots\}$
         Feed-forward: $(a_t^i, m_t^o) = SSoC(s_t^i, M_t)$;
         $M_{buffer} = \varnothing$
         Store message in $M_{buffer}$;
         Sample action using $\epsilon\text{-}greedy\left(a_t^i\right)$
         Execute action;
         Observe reward $r_t^i$;
         Accumulate rewards $R_i = R_i + r_t^i$;
     Observe the next state $s_{t+1}$;
Update network parameters $\theta^1, \dots, \theta^n$ using Eqn.(4);

---

$$\nabla_\theta J_{speak}(\theta) = \nabla_\theta \left[ \log \pi_\theta \left( \boldsymbol{s}_t, \boldsymbol{a}_t^{speak} \right) \left( \sum_{i=t}^{T} r_i - b(\boldsymbol{s}_t, \theta) \right) \right] \tag{3}$$

$$\theta \leftarrow \theta + \lambda \sum_{t=1}^{T-1} \left[ \nabla_\theta J(\theta) + \nabla_\theta J_{speak}(\theta) \right] \tag{4}$$

