# OpenReview forum: "SSoC: Learning Spontaneous and Self-Organizing Communication for Multi-Agent Collaboration"
_ICLR.cc/2019/Conference_

### Official Review · AnonReviewer2 · 2018-11-01
**Paper can be improved by adding more ablation studies around the architecture and doing a more through empirical evaluation.**

**Rating:** 5
**Confidence:** 4

**Review:**

Summary:
This paper expands on the work on 'emergent communication' with 2 innovations:
- The architecture has a separate 'message channel' that processes the incoming and outgoing messages mostly independently of the hidden state of the agent. There are also dedicated architecture elements for the interaction between the hidden state and the message stream.
- The outgoing message is gated with a 'speak' action: only when the agent takes the speak-action at time step t is a message sent out at timestep t+1.

Comments for improvement:
-The paper proposes a rather complicated architecture, with many moving part. In the paper's current form it is extremely hard to see which part of this architecture contribute to the success of the method. A set of ablation studies on the different components would indeed be very helpful.
-Using the word 'thought' to describe the hidden state of the agent is rather distracting.
-Equation (1): This just seems to be the policy gradient term for a factorised action space across 'environment action' and 'communication action'. The only obvious difference is that the policy here is shown to condition on the state representation s_t, rather than on the input. Is that intended?
-The paper suffers from a lot of undefined notation, e.g. the s_t above. Please clarify.
-In Figure 2b) the MCU is shown to produce the action a_t as an output. That seems like a mistake.
-Figure 4): The results seem to be extremely unstable, which is a well known issue for independent learning. Recent work (MADDPG, COMA) has shown that centralised critics can drastically avoid these instabilities and improve final performance. Did you compare against using a centralised critic, V(central state), rather the V(observation)? Also, using a single seed on this kind of unstable learning process renders the results highly non-conclusive.
-In Figure (5), what are the red-arrows? Do these correspond to the actual actions taken by the agents or are they simply annotations? It would be good to see how far the communication range is by comparison. Also, why is there a blob of 'communicating' agents far from the enemy?
-Are different methods in the large scale battle task trained in self-play and then pitched against other methods in a round-robin tournament after training has finished or are they trained against each other?
-In Figure 6 (a), why are average rewards changing over the course of training? I would expect this to be a zero-sum setting in self-play.
-I couldn't find any supplementary material referenced in the text for the details. Instead the paper seems to have another copy of the paper itself attached in the pdf. This makes it hard to evaluate the paper given that few details around training are provided in the main text.

Overall I am concerned that the learning method used in the paper (independent baseline) is known to be unstable and to produce poor results in the multi-agent setting (see COMA and MADDPG). This raises the concern that the communication channel is mostly useful for overcoming the issues introduced from having a decentralised critic.

---

### Official Review · AnonReviewer1 · 2018-11-01

**Rating:** 5
**Confidence:** 3

**Review:**

The paper presents a study on multi-agent communication. The main innovation from previous work is the introduction of an explicit Speak binary action that controls whether or not an agent will emit a message. The proposed model is test on two tasks, multi-camera surveillance and battle tasks with a large number of population.

Overall, this paper is clear (although model details are missing, the authors point at the Appendix, but the Appendix is missing) and the authors compare to a number of baselines. I appreciate the use of multi-agent communication in cases where the number of agents very large,  as this is a very good stress test for current algorithms and can potentially help identifying novel challenges.

My main concern is that in a collaborative setting I don't see why we should expect that occluding information is better than revealing information? Isn't always revealing everything the best strategy? When only 3 cameras, I really cannot think of why a model would get better performance by choosing to not reveal information. Figure 4b somewhat confirms that as there seems to be a lot of variance on the Ssoc. Have you checked how stable are your results across runs? I could believe that occluding information can be beneficial if the number of agents is very big and there is redundancy. But in the limit, not revealing information should only facilitate training --  and indeed this seems to be happening in 6a as Ssoc is learning faster but meanfield is catching up. Could there be an ablation experiment in which everything stays the same in the model but the agents always activate the Speak action? This would answer the question of how crucial this main Speak feature is for Ssoc.

Can you elaborate on this? Moreover, since the Appendix appears to be missing, can you comment on stable results were across runs?

---

### Official Review · AnonReviewer3 · 2018-11-02
**not clear about the originality**

**Rating:** 4
**Confidence:** 3

**Review:**

This paper proposes a spontaneous and self-organizing communication learning scheme in multi-agent RL setup. The problem is interesting. I mainly have one concern regarding its originality.

From a technical perspective, it's not clear to me that there's much novelty in this approach.  I guess it might be the case the focus of the paper is to propose a framework or scheme. However, almost all the ingredients/components are standard.

Regarding clarity, it's not clear to me:
* how the structure from Figure 2 can be reproduced.
* how statistically significant the evaluation results are.

---

### Author Response · Authors · 2018-11-27
**A Revised Version**

In the revised paper, we have updated a complete appendix which is missing in the original submission. Thanks.

---

### Meta-Review · Area_Chair1 · 2018-12-20

**Confidence:** 5
**Recommendation:** Reject

**Metareview:**

The reviewers raised a number of major concerns including the incremental novelty of the proposed and a poor readability of the presented materials (lack of sufficient explanations and discussions). The authors decided to withdraw the paper.